



# Flood hazard mapping and disaster prevention recommendations based on detailed topographical analysis in Khovd City, Western Mongolia

Narangerel Serdyanjiv[1,2], Suzuki Yasuhiro[2], Hasegawa Tomonori[3], Takaich Yoshiyuki[4]

[1]Institute of Geography and Geoecology, Mongolian Academy of Sciences, Ulaanbaatar 15170, Mongolia
[2]Graduate School of Environmental Studies, Nagoya University, Furo-cho, Chikusa-ku, Nagoya, Aichi, 464-8601, Japan
[3]Nippon Koei Urban Space Co., Ltd., Higashi-ku, Nagoya, Aichi, 461-0005, Japan
[4]Nakanihon Air Service Co., Ltd., Toyoyama-cho, Aichi, 480-0202, Japan

*Correspondence to*: Narangerel Serdyanjiv (narangerel.geo@email.com)

**Abstract.** The impacts of climate change manifest heterogeneously across regions, and in Khovd City, a semi-arid area in Western Mongolia, the escalating threat of flooding is evident through the occurrence of 10 flash floods in the last 30 years. The risk zone, encompassing rivers and flash floods, endangers ca. 32,000 residents, with 750-1,800 traditional nomadic dwellings (*ger*s) located on the floodplain of the Buyant River during summer. There is a risk of flash floods in the eastern part of Khovd City from the mountains, while the western part is at a higher risk of flooding from the Buyant River. This paper aims at assessing flood hazards through a detailed topographical survey conducted using an Unmanned Aerial Vehicle (UAV). 15,206 aerial photos were collected in Khovd City using the UAV and measured by Real-Time Kinematic (RTK) on 22 Ground Control Points (GCPs). A Digital Elevation Model (DEM) with a resolution of 2.7 meters was generated from the aerial drone mapping data, enabling surface morphology, hydrological and eight-direction pour point model analysis using a Geographic Information System (GIS). The resulting flood hazard map revealed 4 flood risk areas based on flood flow direction and topographical features. Recommendations for local govern and residents include enhancing flood protection facilities for flood disaster prevention on flood risk zones.

## 1 Introduction

Most cities worldwide are located along the basins of large rivers (Tanaka et al., 2017), and this is also true for cities and provincial centers in Mongolia that traditionaly chose areas with abundant surface water, despite the associated high risk of flooding (Oyunbaatar, 2004). This is of particular concern in climates characterized by short-term but intense rainfall events and significant snowmelt potential. One major indicator of climate change is the distribution and intensity of precipitation in semi-arid regions (Groisman et al., 2005; Sato et al., 2007). Generally, there has been a decrease in mild rain and an increase in heavy and downpouring rain in Mongolia (Goulden et al., 2016; Ministry of Environment and Tourism, 2018). For example, the escalating frequency of sudden floods in semi-arid zones, attributed to global climate change (Sato et al., 2007; Suzuki et al., 2019), poses a particularly high risk to cities in the semi-arid and mountainous regions of Western Mongolia.





In general, Western Mongolia refers to the territory of the western part of Mongolia including 5 provinces (*Aimags*) such as Bayan-Ulgii, Uvs, Zavkhan, Gobi-Altai, and Khovd. Recent flood events in the Bayan-Ulgii *aimag* 2016 and 2018 (Mongolian Red Cross Society, 2019) and Uvs *aimag* 2016 and 2020 have had severe consequences, with casualties and damage to both households and local infrastructure (A Situation of disaster is emerging in Uvs *Aimag* due to flood). A study

by (unpublished, 2023) highlighted that Khovd City of Western Mongolia is highly susceptible to flash floods and river floods, with 10 floods in the last 30 years. Notably, floods in 2018 and 2022 affected more than 230 families, causing damage to homes and belongings. These floods in the mountainous regions of Western Mongolia examplify the significant vulnerability of Western Mongolia to climate change (Ministry of Environment and Tourism, 2018). The increased occurrence and intensity of floods in the region can be explained by the increase of summer rainfall (Goulden et al., 2016)

and melting of glaciers in the Altai Mountains (Pan et al., 2019). The Meteorological Organization of Mongolia has conducted many studies in this field (Ministry of Environment and Tourism, 2018). A recent example of disasterous consequences of changing hydrological condition is the landslide of Mount Tsambagarav (4208m a.s.l) in Western Mongolia in 2021. Zagdsuren.S et al., (2021) linked the flood in Khusni River valley to the melting of glaciers in Mount Tsambagarav. In addition, Narangerel and Suzuki et al (2023) (unpublished, 2023) identified historical floods that occurred in the last 30

years in Khovd City. Using assessments of the damage caused by those floods and surveys among local residents, these authors proposed strategies to improve flood protection facilities and necessary countermeasures future. Continuing this research, the current study emphasizes the need for assessment and mapping of the flood hazard in Khovd City. Recent expansion of residential areas and the increasing number of households (around 1,800) in the flood prone Buyant River valley heighten vulnerability to flooding. With Khovd city's population exceeding 32,000 and expected growth, a detailed

flood hazard assessment becomes crucial for implementing effective prevention and response measures. Mongolia currently lacks natural hazard maps for high-risk areas, making it imperative to evaluate flood hazards through a detailed topographical surveys. Unmanned aerial vehicles (UAV) are used to develop high-precision topographic maps that enable flood risk calculations (Remondino et al., 2012; Restas, 2015; Weintrit et al., 2018; Yang et al., 2022). In Mongolia, UAV equipment finds widespread use in agricultural quality assessment, mining geodesy research, as well as the detailed

determination of natural disaster processes and damage on the ground, along with data collection in emergency areas. However, the utilization of this technology for the development of flood hazard maps is an area where experience is yet to be fully established.

The present study considers the hazard areas of rainfall-derived river floods and flash floods flowing into Khovd City. We consider geomorphological elements such as terrace profiles, floodplains, riverbeds, gullies and depressions to derive

detailed topographical and directional hazard maps. This flood hazard mapping study complements prior research on flood occurrences and historical floods in Khovd City and provides valuable insights for the Administration of Government and Emergency Department of Khovd to protect citizens from flood hazards.





## 2 Area of study

The Khovd *aimag* is located in the Mongolian-Altai mountain range, which constitutes a unique part of the Central Asian mountain system, representing the highest and longest young tectonic mountain systems in the country (Kor Heerema, 2013). Geographically, Khovd *aimag* shares borders with Bayan-Ulgii *aimag* to the west, Uvs aimag to the north, Zavkhan and Govi-Altai *aimag*s to the east, and China to the south (Figure 1). The *aimag* hosts diverse natural zones, with the northeastern part characterized by the lowlands of the Khyargas and Khar-Us lake depressions, desert steppes, and a dry

semi-arid desert subzone. The central region features the mountain steppes of the southern part of the Mongolian Altai, while the southern part is marked by the desert depression of "Baruun Khuurain". The provincial capital, Khovd City, is located in the valley of the Buyant river, a significant river in Western Mongolia (Kor Heerema, 2013; Nara and Battulga, 2019). Khovd City, with its distinctive geographic structure is home to diverse ethnic groups and recognized as one of the historical cities (Nara and Battulga, 2019; Suzuki et al., 2019).

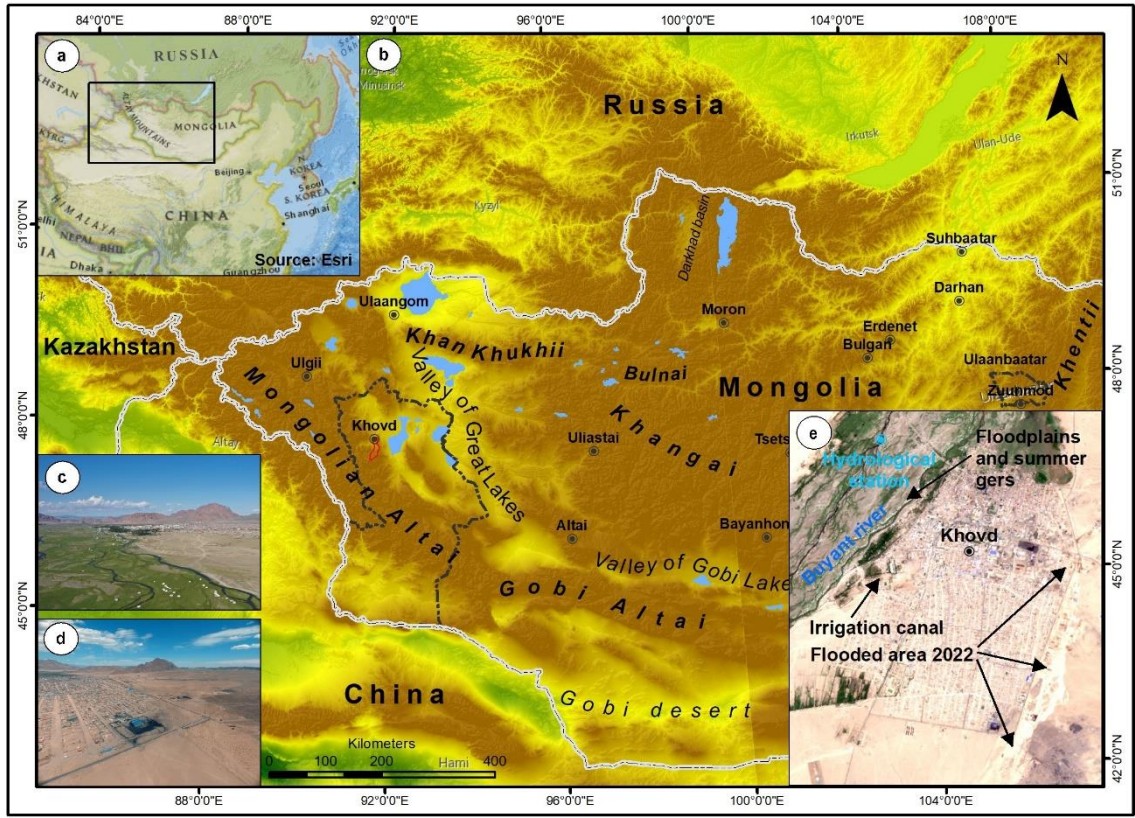

**Figure 1.** The map showing the geographical settings and location of the study area of Khovd *aimag* in Western Mongolia (a, b). It included a western (c) and eastern (d) sides of Khovd City, highlighting features such as the Buyant river floodplain, branches, streams, and the Khovd City area (e-Sentinel 2 satellite image).

Khovd City, located at  1405m a.s.l and 1487 km from Ulaanbaatar capital city, serves as our research site and encomasses

the Jargalant soum with a total area of 70 km2. The Jargalant soum experiences an extremely continental climate,



characterized by dry temperate and mountainous conditions (Beher, 2014; unpublished). The catchment area includes 338 km2, and water accumulated across this region can flow through the eastern part of Khovd City. Positioned at the bottom of a substantial floodplain within a gully, Khovd City is prone to both flash and river floods. The Buyant River valley, passing near Khovd City, is 1.5-2.5 km wide with 2-3 branches and hosts a spring-summer river with yellow water floods. Flash

floods in the Mongolian Altai typically begin in mid-April, reaching their peak in late June. The summer runoff in the rivers of the Mongolian Altai mountain range ends in mid September and during the winter season the river covered by a stable ice cover that lasts for 4-5 months. Due to minimum runoff in winter some large rivers freeze to bottom. The annual maximum runoff ($Q_{max}$) of the Buyant River gradually increased between 1980 and 2009 and intensified significantly over the last decade in  increased summer rainfall. Despite this trend, ca. 750-1800 *ger* and residents continue to spend their summers in

the flood risk zone and are thus exposed to higher risk  (unpublished, 2023).

The Jargalant soum of Khovd *aimag* comprises 12 *baghs* (the lowest-level administrative unit in Mongolia), with around 32 000 people residing in Khovd City. Over the coming years the population will grow and the city will continue to expand. The city itself lies on a relatively plain surface and it is formed by an alluvial fan with a low slope. Notably, a large mountain gully brings flood water from the southeast part of the city, and historical records indicate the occurrence of 10 flash floods

through this gully, resulting in damage to numerous families (Nara and Battulga, 2019; unpublished).

## 3 Methods

### 3.1 Photogrammetry by drone

Recent technological advancements have seen the utilization of Unmanned Aerial Vehicle (UAV) equipment for creating high-precision topographic maps and conducting flood estimation in the study and assessment of natural disasters (Restas,

2015; Weintrit et al., 2018; Yang et al., 2022). Specifically, UAV devices have proven suitable for detailed mapping of small-area flood processes, gaining popularity in flood studies due to their ability to acquire high-resolution images efficiently (Bilașco et al., 2022; Diakakis et al., 2019; Yang et al., 2022). Numerous studies highlighted the use of UAV (Annis et al., 2020; Colomina and Molina, 2014; Feng et al., 2015; Hashemi-Beni et al., 2018; Langhammer et al., 2018; Remondino et al., 2012; Restas, 2015; Schumann et al., 2019; Yang et al., 2022). For instance, research such as Bilasco et

al., (2022) and Annis et al., (2020) showcases the application of UAVs in flood risk assessment.

In August 2019, as part of our field study, we captured 15,206 aerial photographs using the DJI Mavic 2 Pro UAV (Figure 2a) and analyzed them with the Structure from Motion (SfM) program. The specifications of the UAV are detailed in Table 1.


**Figure 2.** Those images show field study works and observations (a, b, c, d and f). There were included drone photo points by UAV (a, f), ground control points /GCP/ by Real-time Kinematic /RTK/ measurement (b, f), interview locations with local residents (c) and geomorphological profile points and flood protection structures (d) etc.

**Table 1.** Specifications of the flight planning and UAV system

| № | Description | Parameter | Value |
|---|-------------|-----------|-------|
| 1 | Camera | Type | DJI Mavic 2 pro |




| | | | |
|---|---|---|---|
| 2 | | Pixel | 5472*3648(20MP) |
| 3 | | Sensor | CMOS (13.2*8.8) |
| 4 | | Shutter speed | 1/1600s |
| 5 | | Pixel size | 2.41 um |
| 6 | | Resolution | 2.82 cm |
| 1 | | Flight altitude | 120 m |
| 2 | | Flight speed | 8-15 m/s |
| 3 | | Overlap | 75% |
| 4 | | Side lap | 64% |
| 5 | | Focal length | 28mm in the 35mm film |
| 6 | Flight | Flight time is approximately | 18 minutes |
| 7 | | Battery capacity | 25-30 minutes |
| 8 | | Number of flight | 53 |
| 9 | | Total flight time | 18 hours |
| 10 | | Number of photos | 15206 |
| 11 | | Coverage area | 1951 ha |
| 1 | RTK, GNSS | Number of GCPs | 22 points |

To ensure the accuracy of Structure from Motion (SfM) models, we validated relative and absolute accuracy using Ground Control Points (GCP) collected through a Real-Time Kinematic (RTK) surveying system. RTK increases the accuracy of GPS signals via a fixed base station that wirelessly emits correcting information to the moving receiver (Schumann et al., 2019). RTK measurements were conducted at 22 GCPs, and the data was processed using the "RTK conv, RTK post, and RTK plot" software, thereby providing detailed location and altitude information for GIS analysis (Figure 2b, e).

The cartographic processing of this study was carried out following the steps shown in Figure 3. We used Digital Elevation Models (DEMs) at two different scales and sources (Alos Palsar and UAV-generated DEMs) with the analytical methods outlined in below.





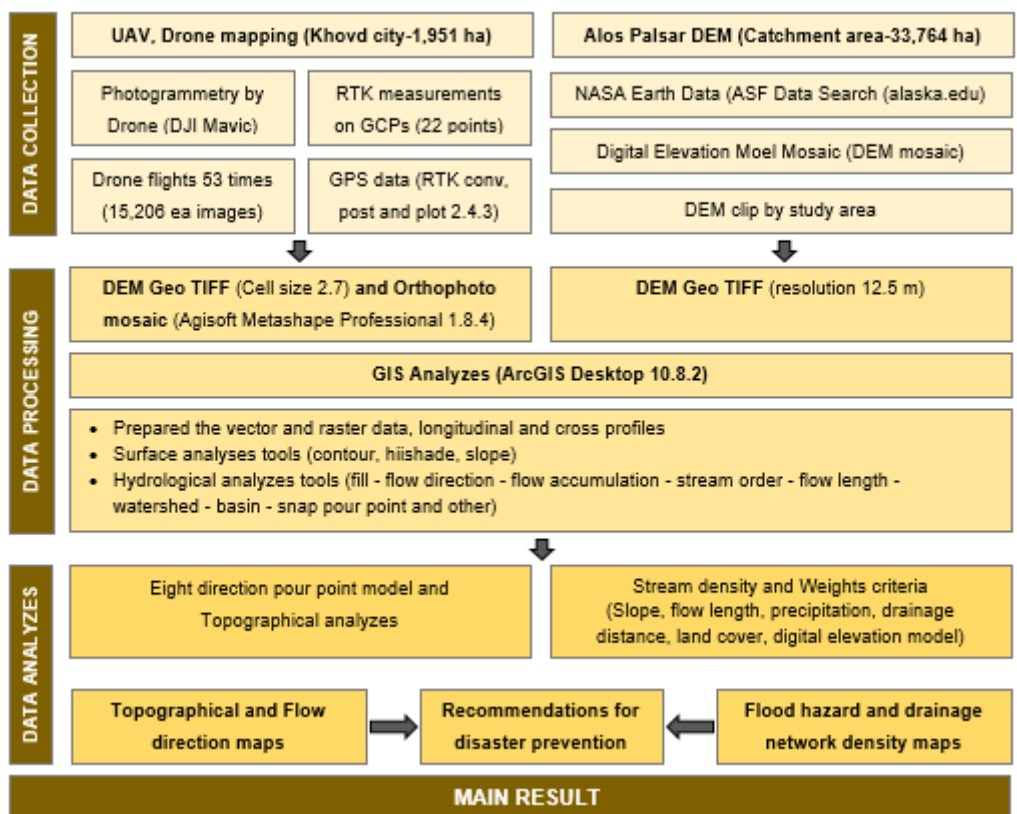

**Figure 3.** Method workflow chart of the study area

As shown in the method workflow, we utilized Agisoft Metashape Professional 1.8.4 software to convert aerial photos into Orthophoto and Digital Elevation Models (OPM and DEM). Agisoft Metashape is an advanced image-based 3D modeling solution designed for professional 3D content creation from still images, offering photogrammetric processing of digital images, including aerial and close-range photography and satellite imagery (2022). The resulting RTK measurements were combined with aerial images to generate a topographical map, creating high-accuracy DEMs. The data, combined with remote sensing, simulation modeling, and other methods, enable a more detailed understanding of flood hazards on a large spatial scale, essential for effective flood management.

**3.2 Topographic and morphometrical survey for estimated flood channel**

We employed Alos Palsar DEM and UAV-generated DEM (Khatami and Khazaei, n.d.; Leitão and de Sousa, 2018; Li, 2014; Turcotte et al., 2001) data to assess topographic and hydrological conditions. Due to the relatively large catchment area, we chose to use the open access Alos Palsar DEM satellite data with a resolution of 12.5 m available from NASA's Earth Data (https://search.asf.alaska.edu/). As shown in the Figure 3, flow direction and stream network density mapping were based on fieldwork and UAV-generated DEM (2.7 m). An orthoimage and a DEM were created using the UAV data



and a flood hazard map was developed for Khovd City by integrating elevation data from two DEM sources for both the catchment area and the city center. Catchment area and stream calculations were performed using ArcGIS Desktop (10.8.2),

with thematic images developed using DEM data at the two scales available in ArcGIS. We employed the "Spatial Analyst tool", "Surface" (Contour, Hiishade, Slope), and "Hydrology" (Fill-Flow-direction-Flow accumulation-Stream order-Flow length-Watershed-Basin-Snap pour point and others) functionalities in the ArcGIS software for flood risk contour and flow direction map development (Eight Direction Pour Point Model) (Castelltort and Simpson, 2006; Khatami and Khazaei, n.d.; Leitão and de Sousa, 2018; Li, 2014; Turcotte et al., 2001; Xu et al., 2001).

Based on the newly developed DEM we generated geomorphological longitudinal section profiles at two points and cross-section profiles at nine points from the Buyant River to the city center. Additionally, geomorphological cross-sections were created in the depression areas using ArcGIS identify flood risk areas. During the field survey, flood embankments, ditches, and water pipes were noted and mapped on-site (Figure 2d). After the flood of 1994, the Khovd administration constructed the initial embankment to block the main channel, redirecting floodwater through the eastern part of the city. The local

government renovated the embankment in 2014, resulting in a 4.12 km protective embankment on the front side of the Khovd City, extending from the airport to the east side of the city, reaching heights of 0.6-1.8 m and widths of 0.8-2.8 m. Furthermore, a 4.24 km drainage channel/ditch, 0.8-2.0 m deep and 1-3 m wide (shallow and narrow in some parts), connects with the embankment. Additionally, a 2.24 km drainage ditch was constructed on the opposite side of the road, serving as a supplementary protection measure in some areas. Around the city center, ca. 20 drainage structures with

diameters ranging from 1-1.5 m are mostly located on the east side of the Khovd city's flood channel (Figure 2d).

### 3.3 Topographic numerical information analysis

Topography plays a crucial role in cartographic science, providing a quantitative assessment of the shape and landform of the surface relief, commonly depicted through contour lines. The digitalization of topographic maps is of great importance for the analysis of topographic numerical information (Govedarica and Borisov, 2011; Ruhoff et al., 2011).

Here, wedeveloped topographical maps utilizing UAV-generated DEMs in Khovd City. To analyze flow direction, we employed the "Eight Direction Pour Point Model," resulting in the creation of flood hazard maps for Khovd City. The flow method necessitates two high-resolution topographic datasets – a flow direction map and a surface elevation map – at the same resolution to generate a lower-resolved river network map and supplementary maps of river network parameters (Yamazaki et al., 2009). The eight-direction flow direction coding was applied by considering that the stream flows from the

center cell to its eight neighboring cells, and assigning a number to each of the eight neighboring cells based on the direction of flow. We determined the drainage structure of a watershed using DEMs and a digital river network. While it is relatively straightforward to estimate natural flow direction from DEMs, deriving accurate flow direction is challenging for artificially modified river channels and human-made structures based on DEMs alone. Our topographic map enabled the creation of cross sections and longitudinal profiles along the Buyant River. In addition, we conducted an analysis of terraces in areas





where water may enter Khovd City determined the flow direction of rivers and flash floods and examined water
accumulation depressions in the center of Khovd City.

## 4 Results

### 4.1 Topographic maps and flow direction analysis

According to the derived topographical maps of Khovd City, it locates in an alluvial fan with an elevation ranging from 1344

to 1410 a.s.l and very shallow surface slopes of ca. 2-4 degrees. The alluvial fan is characterised by a total elevation
difference of about 60 m with the eastern part of the city area meeting the foothills of the Ulaan (2026.4m) and Tshair
(1642.0m) of the mountains and the western part directly connecting to the floodplain of the Buyant River valley (Figure 4).
The northern section of the study area region converges with the Buyant River wetlands and represent the lowest area where
all runoff from higher elevations converges. Although it is possible to surface water colleted from catchment area and gentle

slope of the surface, the flow intensity is relatively slow.



**Figure 4.** The map shows the topographical conditions and flow directions of the Khovd City, featuring a total of 11 profiles. These profiles include two longitudinal profiles (PR-1, PR-2 along the red line) following the course of the Buyant River and an irrigation canal. Additionally, there are nine cross profiles (PR-3 - PR-9 along the black line, and PR-10, PR-11 along the blue line) encompassing the Buyant River and two depressions.

Regarding the surface water catchment area, the primary flow was along the flood embankment and drainage ditches in the eastern part of Khovd City. This area has experienced numerous floods in the past, affecting households in the Baatarkhairkhan, Bichigt, Naran, and Rashaant *bagh*s. The density of the flood drainage network in this area is the highest, ca. 6.5 km/km². The flow direction in the central area of Khovd City was calculated using the DEM based UAV. The analysis, based on a cell size of 2.7 m grouped into 120 m cells, identified a total of 980 cells with distinct flow directions. Examining the flow directions as a percentage in the study area, we find that 1.3% flows southward, 56.5% northward, 4.9%




westward, 3.6% eastward, 4.0% southwestward, 0.4% southeastward, 11.3% northwestward, and 18.0% northeastward. According to the Eight Direction Pour Point Model the majority, 85.8%, are directed south to north, northwest, and northeast (Figure 4). This result indicates that, in the event of rainfall in Khovd City, most runoff will move towards the wetland in the

low-lying part of the Buyant River valley. The flow direction map reveals that the high risk flow originates from flash floods from the east side of Khovd City. Our study identifies two depressions in the center of Khovd City, and the flow direction can enter these depressions from all sides (Figure 4, Figure 5). Flood water flow trends to accumulate in these depressions. The topographical map also illustrates that floodwater can move towards the center through the irrigation canal drawn from the Buyant river, a topic that will be discussed in detail below.

## 4.2 Topographic depressions and flood water accumulation

Climatic factors play a significant role in the formation of micro-landforms. Especially in the low-lying areas of wetland river valleys, surface depressions commonly due to the thaw of permafrost, enhanced by global warming (Jones, 1993; Karjalainen et al., 2020; Twidale, 2002). Yamkhin et al. (2022) estimated that permafrost covers 29.3% of Mongolia's total territory with Khovd City located in the seasonal frost zone (Pan et al., 2019; Yamkhin et al., 2022).

The topographical analysis conducted in this study identified two depressions in the center of Khovd City, specifically in the areas of Buyant, Naran, and Rashaant *baghs*, as shown in Figure 4 and Figure 5 (profile 10 and 11). The profile PR-10 in the Rashaant *Bagh* area indicates a surface elevation of about 1350 m a.s.l. with a depression depth of 1.5-1.7 m. Similarly, the profile PR-11 in the Buyant *Bagh* area exhibits an elevation of about 1362 m a.s.l. with a depression depth of 2-2.5 m.

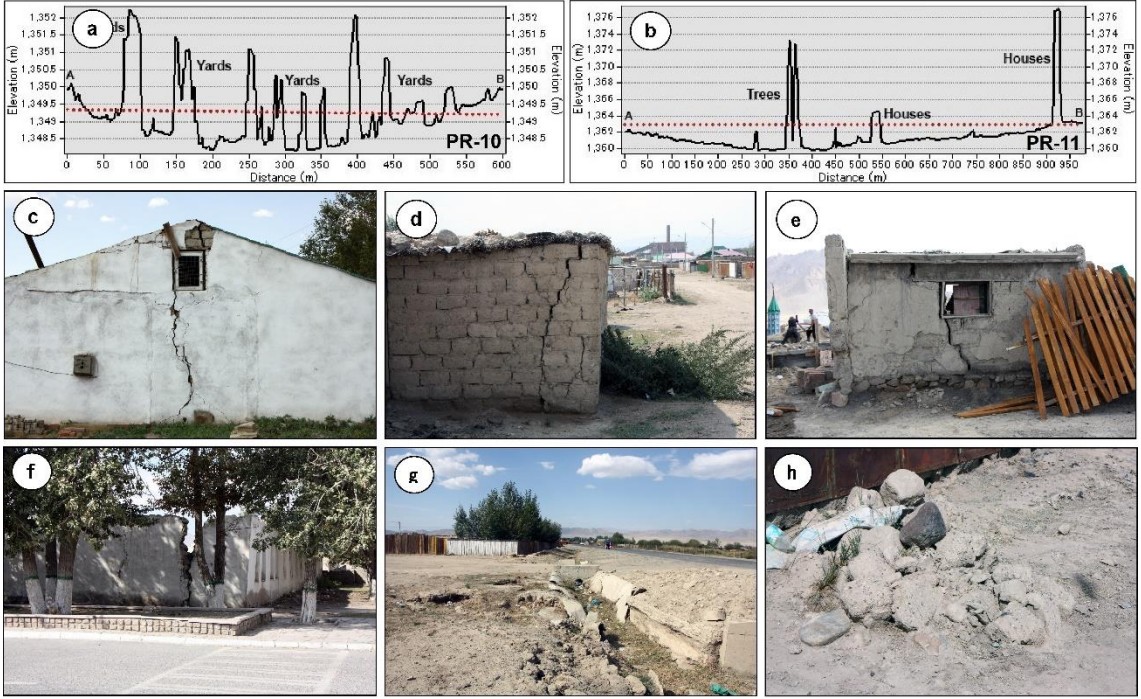





**Figure 5.** The depression profiles PR-10 (a) and PR-11 (b). Broken buildings (c, d, e, f) and permafrost sediment (g, h) on the depressions in Khovd City.

Although the seasonally frozen ground is crucial for the water regime of the Buyant River, it is highly susceptible to global warming and human activity. In the Khovd area, the soil typically freezes at the end of October and begins to thaw in mid April (Kor Heerema, 2013). The proximity of the soil water level to surface in this region may have contributed to the development of depression microrelief through frost action (thaw/freeze cycles). The high soil water table is confirmed by private wells in this area. The observed collapse or destruction of houses and buildings (Figure 5a, c, d, e, f), in the Naran and Rashaant *bagh* areas (profile PR-10) and sediment analysis in these areas suggest that these depressions likely result from permafrost degradation over the last decades (Figure 5g, h) (Walther and Kamp, 2023). There are no comprehensive studies on the origin and ground freeze characteristics of these depressions, so our interpretation is based on the available images. Furthermore, if flash and river floodwaters enter the city center, the low-lying areas and depressions in sections PR-10 and PR-11 are more severely affected with more sustaied damage. The topographical and flow direction maps (Figure 4, Figure 5a, b) illustrate that rain and floodwater trend to accumulate in these areas. According to residents, many families located in the depression (profile PR-10) experienced flooding in July 2018.

**4.3 Impact of Buyant River floods**

The geographical location of most provincial centers in Mongolia, particularly those in semi-arid of Western Mongolia, places them in the valleys of larger rivers originating in the Altai Mountain ranges. In Khovd City, despite a general decrease in precipitation in last decades, our previous research indicates that heavy sudden rainfall thawing permafrost and melting glaciers in the Altai Mountains contribute to an increase in the warm season runoff and flood intensity of the Buyant River (Kor Heerema, 2013; Pan et al., 2019; Yasuhiro et al., 2019; Zagdsuren.S et al., 2021). The rising water levels and increased flow in larger rivers pose a serve flood threat to nearby settlements and cities. Large rivers in Mongolia are known to flood rapidly with a lag of 1-3 days after rainfall, causing a rapid rise in river water levels (Oyunbaatar, 2004).

The Buyant River is sustained by spring yellow water (snow meltwater), summer rains, and warm season meltwater from permafrost and glaciers in the Altai Mountains. Spring yellow water floods typically start from late April to mid-May, with the peak observed by the end of June in the Altai Mountains. The summer rainfall flood in the middle of the Mongolian Altai mountain range usually ends in mid-September. During the winter, the river experiences stable ice cover lasting 4-5 months, with other large rivers freezing to the bottom. In years of high water, the Buyant River's water level rises above the long-term average, for example, reaching 336 cm in August 2019, which is 22 cm higher than the long-term average (Climate change around Khovd city). The Buyant River has 2 branches as it flows through Khovd City, creating additional tributaries during floods that significantly raise water levels. Moreover, the westernmost tributary runs along the city's border, with local farmers using 3 canals to irrigate their vegetable areas. These canals contribute to water flow into the city center, increasing the flood risk in households. In case of prolonged rains that lead to Buyant River flooding, not only houses along the river valley but also in the city center could be affected by the flood water (Figure 4, Figure 6).


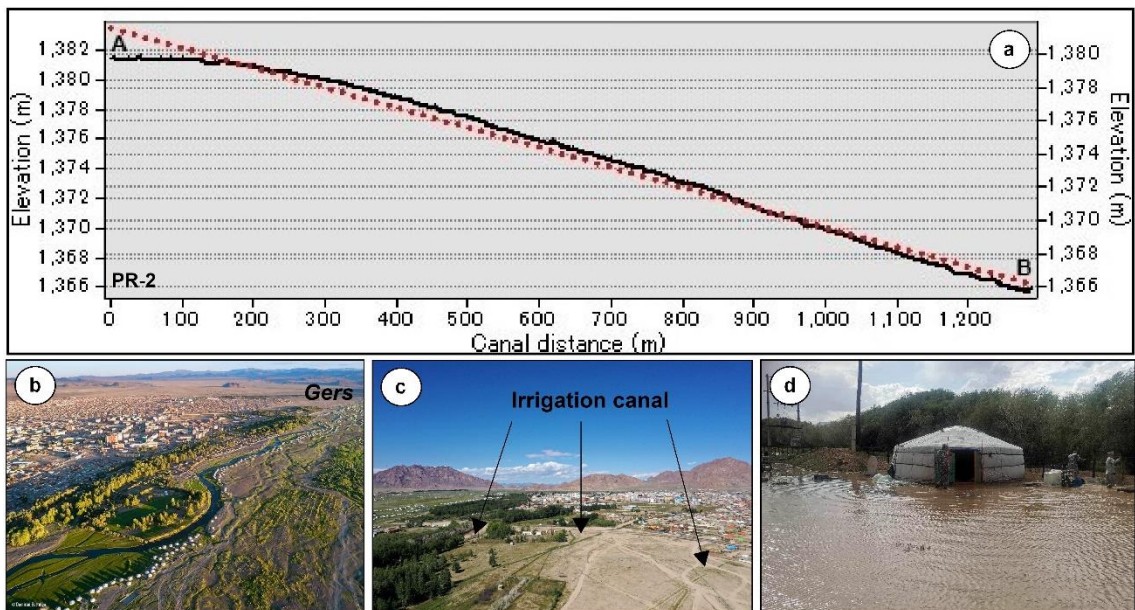

**Figure 6.** The images depict the longitudinal profile PR-2 of a vegetable irrigation canal (a). The Buyant River valley and summer *gers*
near Khovd City are shown in (b). Additionally, there are artificial irrigation canals/channels leading into the city, as seen in images (c, d).

Local farmers have constructed three irrigation canals along the surface of the Buyant River that lead to the center of the
city, this is visible in profile PR-2. The longitudinal profile in (Figure 6a, c) indicates a significant elevation change that
provides a pathway for flood water flow into the center of Khovd City along through these irrigation canals. The longitudinal
section of the Buyant River west of the city, represented by profile PR-1 is characterised by a relative height difference of 50
meters from 1395 to 1345 m a.s.l with a river gradient of 0.01 (Figure 7a). In the initial segment of the longitudinal section
and the upper part of the river, terraces no. 1 and 2 are found in the river valley, with terrace heights ranging from about 4-6
m on profiles PR-3 and PR-4 (Figure 7b, c). The relatively high elevation of these terraces constrains the flood waters from
the Buyant River in this area and prevents direct flood flow into the city of Khovd (Figure 8a, b).



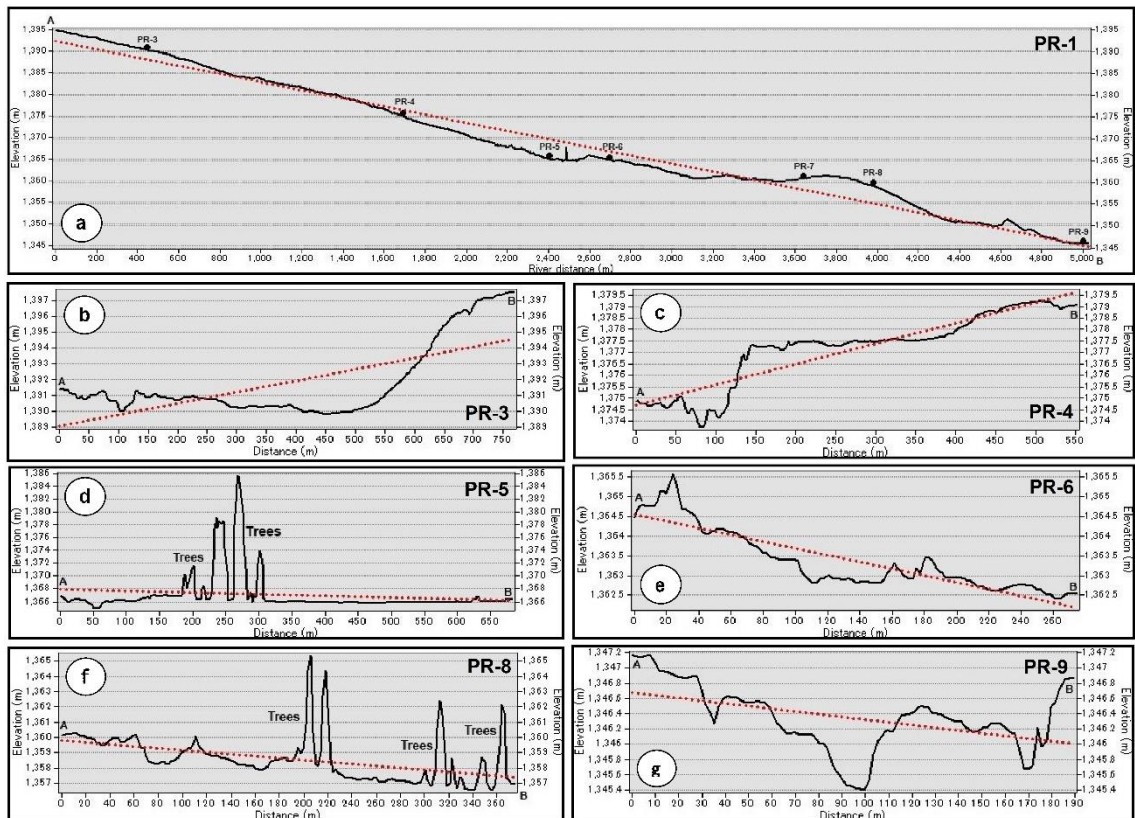

**Figure 7.** The longitudinal (a) and cross profiles (b, c, d, e, f, g) of Buyant River near Khovd city area.

However, at the level of the cross-section PR-5 (Figure 7d), the elevation of the terrace subsides and disappears. From this point onward, the surface level of the center of the city decreases by 1-3 m relative to the base of the river valley bottom (1367-1366 m a.s.l) thus providing conditions for floodwater to flow towards Khovd City (Figure 7e, f, g and Figure 8c). Ancient terraces of the river valley are not found below this area.




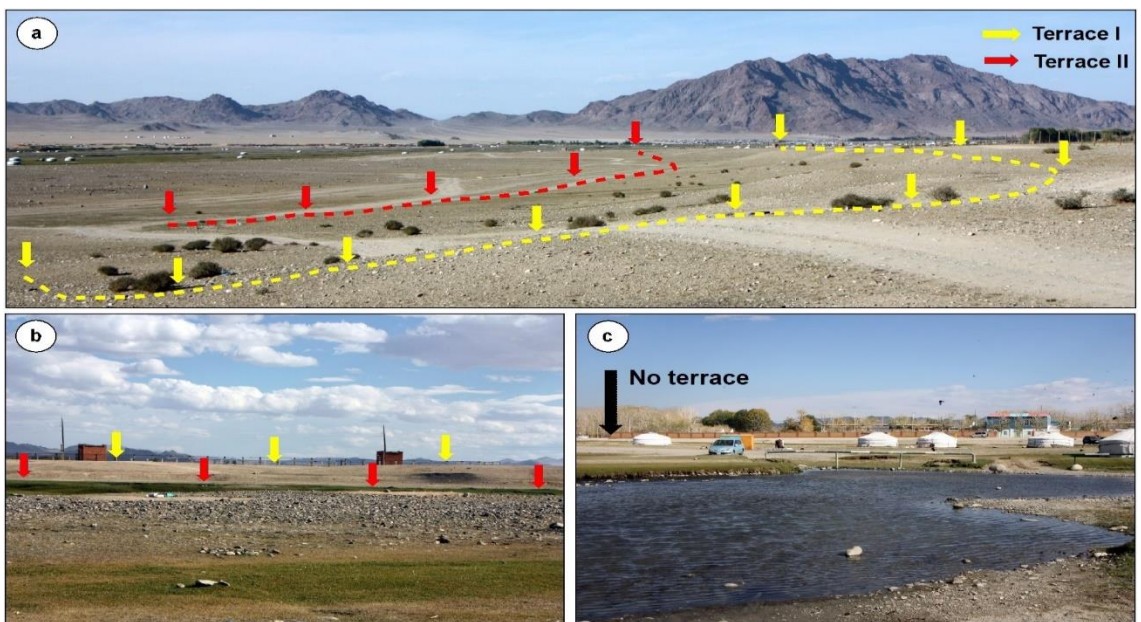


**Figure 8.** Elevation changes of the Buyant river terraces at PR-4 (Figure 7b) and PR-5 (Figure 7d).

## 5 Discussion

### 5.1 Potential flood hazard impacts according to topographical analysis

Khovd City currently sustains a population of approximately 32,000 residents, with a density of 440 people/km² (Khovd
Statistical Office, 2020). The population density in the provincial center increased notably in recent years, despite the fact
that the city faces a significant risk of natural disasters and in particular floods triggered by climate change (Nara and
Battulga, 2019; Yasuhiro et al., 2019). In response to this risk, we developed a hazard map depicting flow directions using
our new a topographic map. Our created the flood hazard map identified four hazardous and vulnerable areas within Khovd
City (Figure 9) including:

270       1. The Buyant River which bifurcates into 2-3 branches near Khovd City with additional tributaries during flood
events which elevate water levels significantly. The summer *ger*s located on the floodplains between the river's branches
face a high risk of inundation. The expanding residential area of Khovd City, with currently ca. 1,800 households in the
flood hazard area, aggravates the vulnerability to flooding. Recent studies noted an increase in the Buyant River's maximum
runoff (Qmax), indicating a heightened risk of rain floods  (unpublished, 2023).

275       2. The westernmost tributary, running along the city border, contributes to the local irrigation network that supplies
water for vegetable produce through 2-3 canals. These canals are linked to an irrigation channel drawn from the Buyant
River and pose a potential threat to the city center. Flood risk extends to households in Buyant, Tsambagarav, and Jargalant
*bagh*s, especially those in the Buyant *bagh*'s depression (see profile PR-11).
3. Flow direction analysis in Khovd City center reveals two depressions with increased flood risk. The depressions in
Naran and Rashaant *bagh*s consistently experience flooding. Flash floods entering the city center may affect the depression highlighted in section PR-10 more severely resulting in greater damage (Figure 9).

4. According to the Eight direction pour point model most water flows nothward which mean that in the event of rainfall in Khovd City most of the flow directed towards the wetland and lower parts of Buyant River valley. With the wide catchment area and narrow outlet and water culvert pipes, flash floods originating from the dry valley and gullies of "Dan
Usnii Khundii" originating from "Khar Tsohio" in the southeastern part of Khovd City, pose a high risk. We find a higher likelihood of overflowing floodwaters, endangering households in Baatharkhairkhan, Bichigt, Rashaant, and Naran *bagh*s. Notably, this area hosts the highest density of the flood drainage network, measured at 6.5 km/km².

**Figure 9.** The maps are showing flood hazard places in Khovd City. And those maps show a flood flow direction and depressions of the
study area.





In Khovd City, 72% of the population resides in *ger*s (nomadic dwellings), 26.5% in houses, 1.2% in apartments, 0.2% in independent comfortable houses, and 0.1% in public housing (Khovd Statistical Office, 2020). Notably, residents living in *ger*s are particularly susceptible to higher risks and damages because it is very vulnerable to flood flowing and easily
penetrated by flood water. Past experiences of flash floods in the city indicate that individuals residing in cities have endured more substantial harm and property damage (unpublished, 2023). In this regard, local elders says about the flood of 1994 when many *gers* were flooded. Consequently, there is a critical need to mitigate the risk of flooding in Khovd City as in other cities and implement measures to protect and prevent the adverse impacts of floods on its residents.

### 5.2 Recommendations for disaster prevention

Building upon the topographical analysis conducted, we propose the following general flood prevention measures:

1. Urgent action is required for flood prevention, protection, and disaster management, with special emphasis on the four identified points in the new flood hazard map. Key recommendations include i) enhancing flood protection facilities, ii) regular warnings to residents in flood-prone regions or river valleys, and iii) the implementation of local disaster management initiatives. Additionally, efforts should be made to improve the quality of weather prevention information,
organize training sessions, and raise awareness about natural disasters among the citizenry.

2. Given the increasing frequency of floods in the Buyant River in recent decades and the likelihood of further aggravation associated with ongoing climate change local authorities should take proactive measures to ensure the safety of the 1800 families residing in vulnerable areas of the river floodplain. It is paramount to establish a comprehensive rescue preparedness plan for sudden natural disasters and particularly floods. Distributing informative materials to raise awareness
and equip local administrative bodies and residents to protect themselves from floods is crucial.

3. Long-term changes in local rainfall pattern pose a risk of flash floods and the potential inundation of the two depressions in the city center by the Buyant River. It is imperative to safeguard households in these areas and to construct flood protection structures based on robust flood protection design. A detailed study of the seasonal frozen ground and permafrost dynamics in these depressions should be conducted and measures informed by this study should be urgently
implemented.

4. The flood protection structures erected in 1994 and 2014 have experienced serious deterioration and require urgent reconstruction and improvement.

### 6 Conclusions

This paper presents a detailed flood hazard assessment of Khovd City based on high-resolution topographical surveys
conducted using an unmanned aerial vehicle. UAVs prove highly suitable for monitoring river morphology and flood drainage in small areas, providing very high spatial resolution orthomosaics and DEMs essential for calculating




morphological changes. The paper emphasizes the need to protect Khovd City from future floods, given its location in the semi-arid region, serving as a provincial center in Western Mongolia.

We discuss floodwater flow direction and identify two depressions within the city, primarily oriented from south to north

that are particulary vulnerable to flood risk. While the origins and permafrost conditions of these depressions require more detailed investigation, we can gain insights into flood risk from UAV-based contour maps and photographs. In case of flash floods or river inundation in the city center, the depressions of sections PR-10 and PR-11 are expected to suffer more significant damage. Four flood-prone areas were identified in Khovd City, which we discuss to formulate flood prevention recommendations. The Khovd City administration should provide flood disaster management knowledge for residents and

improve flood protection engineering facilities in climate change adaptation and disaster resilience.



*Data availability.* Data will be made available upon request.

*Supplement.* The supplement related to this article is available online at:

*Author contributions.* NS conducted field research, analyzed data, and wrote the manuscript. SY planned and supervised the research project. HT and TY provided drone photos. All authors read and approved the final manuscript.

*Competing interest.* The authors have no relevant financial or non-financial interests to disclose. The authors declare that they have no known competing financial interests or personal relationships that could have appeared to influence the work reported in this paper.

*Acknowledgements.* Appreciation is extended to the staff of Khovd NEMA and the government office of Jargalant soum in Khovd *aimag*
for their support during fieldwork. We thank Associate Professor Sebastian Breitenbach of Northumbria University for reviewing the English. Special thanks are given to the Asian Satellite Campuses Institute (ASCI) Program of the Graduate School of Environmental Studies (GSES) at Nagoya University for their support in the doctoral program. The authors also thank all stakeholders involved in the study for their cooperation.

*Financial support.* The authors express gratitude for financial support from JICA Grass-roots Joint Project (JPJSBP120209913), JSPS
Bilateral Joint Research Projects (JPJSBP120209913), JSPS KAKENHI (16H05645), and JSPS KAKENHI (18K18543).

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
