# Peer review of "Flood hazard mapping and disaster prevention recommendations based on detailed topographical analysis in Khovd City, Western Mongolia"

_Natural Hazards and Earth System Sciences, 2024_

## Author Comment (AC2)

**Review to Flood hazard mapping and disaster prevention recommendations based on detailed topographical analysis in Khovd City, Western Mongolia, Author(s): Narangerel Serdyanjiv et al., MS No.: nhess-2024-91**

**RC1: Comment on nhess-2024-91, Anonymous referee #1, 18 Jul 2024**

First of all, many thanks for all your valuable comments and suggestions. Please see below for changes based on your comments.

**General comments:**

In the study "Flood hazard mapping and disaster prevention recommendations based on detailed topographical analysis in Khovd City, Western Mongolia", authors present rainfall runoff maps based on digital elevation data derived from UAV flights.

**1.1.** Unfortunately, the study lacks novelty and evaluations lack scientific quality. Results are of interest to local decision-makers but do not contribute scientifically enough on their own. Other than stated in the introduction, riverine flooding is not accounted for, but only surface runoff.

Yes, you're right, this study is more than a theory, it describes the natural disaster risk situation that the region is facing and aims to provide scientific recommendations to policy makers in the region to prevent floods. We mainly focused on the development of flood hazard mapping (surface water runoff and flash flooding) based on UAV. Changes have been made to the abstract section of the manuscript based on your comments. It Included:

***Abstract.*** *The impacts of climate change manifest heterogeneously across regions, and in Khovd City, a semi-arid area in Western Mongolia, the escalating threat of flooding is evident through the occurrence of 10 flash floods in the last 30 years. The risk zone, encompassing rivers and flash floods, endangers ca. 32,000 residents, with 750-1,800 traditional nomadic dwellings (gers) located on the floodplain of the Buyant River during summer. Furthermore, prolonged rains pose a flash flood risk to households in the center of the province. However, different disaster prevention measures are required compared to those for river flooding in humid regions such as Japan and Southeast Asian countries. In Khovd, residential areas are limited, and land use is not highly dense. In addition, since flood water levels are not high, knowledge of the location and direction of flood flow paths and places where water is likely to be collected in advance is essential for disaster risk reduction. Under these conditions, mapping using detailed DEM and identifying the extent of past floods using satellite images are important. We measured by Real-Time Kinematic (RTK) on 22 Ground Control Points GCPs and collected 15,206 aerial photos for drone mapping under Unmanned Aerial Vehicle (UAV) in the Khovd City. The purpose of this paper is to use Khovd as an example to create a hazard map based on topographical analysis of detailed DEM, and to discuss a methodology for using this map to help with flood disaster prevention in remote areas. The resulting flood hazard map revealed 4 flood risk areas based on flood flow direction and topographical features.*

**1.2.** Further, the areas at risk are not specified or classified, which would be of interest to decision-makers, and first responders, and support the adaption process.

Using topographical and surface flow direction maps (Figure 4), areas at risk of flooding are identified and explained in Section 5.1 with Figure 9.

**1.3.** I would advise you to continue and expand your analysis, because it seems, that additional data and information are available, but were not used.

Changes have been made to this manuscript and are marked in red.

**Specific comments:**

**1.4.** An evaluation scheme for the risk areas including classification would be helpful to clarify the identified risk areas.

In the 4th section of the manuscript, when determining flood risk areas, the risk areas were identified as a result of analysis of topographic condition and flow direction analysis and geomorphic profiles. Therefore we checked the risk areas on the places.

**1.5.** As further field data exist (geological, morphological, hydrological?, maybe critical infrastructure?), I recommend conducting further analysis.

**1.6.** Could the study benefit from evaluating and including the local interviews (shown in Figure 2)? If not, please explain, why the locations of interviews are shown in Figure 2.

We collected questionnaires from 54 local residents during the field mapping study and double-checked flood risk areas. Added a small sentence about it on 3.1.section. It Included:

*In addition, during the field mapping step, the flood hazard areas were checked under questionnaires from local residents.*

**1.7.** Would a multi-hazard evaluation be possible, which includes reconstructed or modelled fluvial inundation?

It is good to think about it

**1.8.** Line 135ff: it is unclear to me, when the Alos Palsar DEM satellite data and when the UAV-generated DEM was used in your study, or how you combined the two DEMs

You're right, the Alos Palsar DEM was not used in this study to a particularly high importance, so the all means was removed about Alos Palsar DEM from the manuscript altogether. It was my mistake sorry.

**1.9.** Line 174: is it possible to show the alluvial fan on a map?

We can see on the Figure 1 c, d, e, Figure 2 d and Figure 9

**1.10.** Line 180: please specify "relatively slow"; e.g., please add significant values

Changed the word

**1.11.** Figure 4: could you additionally show inundation depths?

In the semi-arid region like Khovd, where the risk of flooding increases due to global warming, it is necessary to quickly identify areas where water is likely to collect due to topographical factors and notify residents of the risk. The inundation depth is generally not large, about several tens of centimeters. In humid regions, flood hazard maps that clarify the inundation depth through runoff analysis simulations are useful, houwever creating such hazard maps requires both budget and time. The purpose of this paper is to propose a simple preliminary flood hazard map creation method in such an urgent situation.

**1.12.** Line 190: how is the grouping of the cells done, and for what purpose?

In detailing the DEM we created from the drone mapping and aerial images , we calculated the percentage of each cell showing its own flow direction into the atribute database on GIS. This is important in determining where surface runoff may accumulate and pose a hazard.

**1.13.** Is the bed or the water surface area of ditches and rivers shown in the DEM? If not, does the study benefit from embossing the bed level to the DEM?

Yes, we showed the drainage ditch and river in the DEM as Figure 2 and Figure 9.

**1.14.** How are flow directions further evaluated to overall evaluate the risk posed by flash floods?

Additional information on this is provided in Section 3.3 by red colour

*To analyze flow direction, we employed the "Eight Direction Pour Point Model," resulting in the creation of flood hazard maps for Khovd City. The flow method necessitates two high-resolution topographic datasets – a flow direction map and a surface elevation map – at the same resolution to generate a lower-resolved river network map and supplementary maps of river network parameters (Yamazaki et al., 2009).*

The eight-direction (D8) flow direction coding was applied by considering that the stream flows from the center cell to its eight neighboring cells, and assigning a number to each of the eight neighboring cells based on the direction of flow.  For an input D8 flow direction raster, a cell is considered to have an

*undefined flow direction if its value in the flow direction raster is anything other than 1 (eastward), 2 (southeastward), 4 (southward), 8 (southwestward), 16 (westward), 32 (northwestward), 64 (northward), or 128 (northeastward) . Output cells with a high flow accumulation are areas of concentrated flow and can be used to identify stream channels.*

**1.15.** 98-105: please consider moving the background information to the introduction/background

Yes sure, it moved to the introduction/background

**1.15.** Line 147-171: please move to the description of the study area

Yes sure, it moved to the the study area

**1.16.** Lines 201-204: please consider moving the background information to the introduction/background

**1.17.** Lines 212-223: Please provide further information on the sedimentological data you mention (methods, results, consideration in the study). The section belongs to the discussion. Please provide the climatologic data, you are referring to.

**1.18.** Lines 225 to 242 & 246-253: The section covers parts, that belong to the introduction, study area, and discussion. Please provide the corresponding results, and move sections.

In this manuscript, I wanted to provide an understanding of the risk of flash floods in the dry valley and gullies of Khovd city and the flood risk of the Buyant River. These 2 flood conditions are separate understands and conditions. In other words, this city is located in the middle of the flash flood condition and the river flood condition.

**1.19.** Line 268: it is unclear how areas were determined. What are uncertainties? How much higher is the risk compared to other areas?

Changed the sentence

*Based on the results of this study, we created a flood hazard map using the information on topographical conditions, flow direction, residential settlements area distribution of households, and previously flooded areas and identified four hazardous and vulnerable areas within Khovd City (**Error! Reference source not found.**) including*:

**1.20.** Fig 9: how does the hazardous flow direction differ from the general flow direction?

We wanted to show the local residents that there is a possibility of flooding in this area and accumulation in the depressions in by hazardous flow direction. In the 4 areas shown in Figure 9, many householders have been affected by floods in recent years.

**1.21.** Recommendations are too general. Please link them more closely to your results

Yes sure, we added new sentences in 5.2 section as below:

Geomorphological hazard mapping of this area can alert residents to potential unknown risks. Specifically, it solves the following problems: 1) Because the area is semi-arid and there is no running water, residents do not recognize the risk of flooding. 2) Because the terrain is generally flat, residents do not know where the water will flow during floods. 3) After an artificial embankment was built in the 1990s, the flood flow path changed, but residents cannot predict what will happen if the embankment breaks. 4) The Buyant River flows at a low level on the west side of the city, but the residents of Khovd have never experienced floods there historically, so they are completely unaware of the risk of flooding.

**Technical corrections:**

**1.22.** Could you please specify the unpublished literature (e.g., cite the publications as "under review", if applicable)

Corrected

**1.23.** Figure 1: Location of pictures is not marked on the map

We provided figure's explains, is it necessary to mark on the map?

**1.24.** Line 80 ff: km2 must be km²

Corrected

**1.25.** Line 91: please check the line break at the end.

Corrected
**1.26.** Figure 2: Please change /xx/ to (xx)
Corrected
**1.27.** Table 1: the information seems a bit excessive. Could you remove parts of it, by referring to the UAV you used?
Which one to exclude?
**1.28.** Line 141: Hiishade must be Hillshire
Corrected …Hillshade
**1.29.** Line 141 ff: (Fill-Flow-direction-Flow accumulation-Stream order-Flow length-Watershed-Basin-Snap pour point and others) commas are missing
Corrected
**1.30.** Figure 5 and Figure 7: the dashed red line is missing in the legend
Added explain sentence in the figure desription

*The red dashed line shows the profile slope.*

---

## Author Comment (AC3)

**Review to Flood hazard mapping and disaster prevention recommendations based on detailed topographical analysis in Khovd City, Western Mongolia, Author(s): Narangerel Serdyanjiv et al., MS No.: nhess-2024-91**

**RC2: Comment on nhess-2024-91, Anonymous referee #2, 30 Jul 2024**

First of all, many thanks for all your valuable comments and suggestions. Please see below for changes based on your comments.

**General comments:**

The authors describe a trend of increasing flood intensity and occurrence in Western Mongolia and the lack of natural hazard maps making developing and implementing mitigation strategies difficult.

In response, they derived, using UAVs and Alos Palsar DEMs flood hazard maps based on geomorphological elements and identified 4 flood risk areas.

**2.1.** However, flood hazard mapping which may be lacking in Western Mongolia, is not novel, nor is UAV utilization to develop hazard maps.

**2.2.** It is clear that the application of results from the presented paper is important to the local population and practitioners and in further developing Mongolian hazard planning but does not contribute significantly to the field of natural hazard assessment and related processes.

In the semi-arid region like Khovd, where the risk of flooding increases due to global warming, it is necessary to quickly identify areas where water is likely to collect due to topographical factors and notify residents of the risk. The purpose of this paper is to propose a simple preliminary flood hazard map creation method in such an urgent situation. Geomorphological hazard mapping of this area can alert residents to potential unknown risks. Specifically, it solves the following problems:

1) Because the area is semi-arid and there is no running water, residents do not recognize the risk of flooding.

2) Because the terrain is generally flat, residents do not know where the water will flow during floods.

3) After an artificial embankment was built in the 1990s, the flood flow path changed, but residents cannot predict what will happen if the embankment breaks.

4) The Buyant River flows at a low level on the west side of the city, but the residents of Khovd have never experienced floods there historically, so they are completely unaware of the risk of flooding.

**2.3.** A potential suggestion that would, in my opinion, make the results more impactful is to use the derived maps (DEMs), inventories of critical infrastructure, and relevant hydrological data to expand upon the current understanding of flood processes in the region with 1d or 2d (as suggested in line 130) numerical modeling to better delineate hazard zones as they relate to discharge scenarios, and the potential hazard reductions in response to deployed mitigative measures.

We mainly focused on the development of flood hazard mapping (surface water runoff and flash flooding) based on UAV. In the semi-arid region like Khovd, where the risk of flooding increases due to global warming, it is necessary to quickly identify areas where water is likely to collect due to topographical factors and notify residents of the risk. The purpose of this paper is to propose a simple preliminary flood hazard map creation method in such an urgent situation.

**2.4.** Therefore, I opine that additional analysis and synthesis are required which would ideally include models that show inundation zones (and relevant hydraulic variables) and deal with anticipated changes in the rainfall regime which would greatly improve the quality of the manuscript while making the results more impactful to a broader readership.

This mean is covered in our previous article, which is currently under review. Thanks for your valuable advice.

**Specific comments:**

**2.5.** The introduction requires refinement and more focus, such as elaborating on the strategies Narangerel and Suzuki et al (2023) developed to improve flood protection and if the current growth and expansion follow these guidelines.

In this regard, additional changes were made to the abstract and introduction of the manuscript.

**2.6.** It would be interesting to know how the presented work aims to improve or modify what is currently available or practiced in terms of hazard response.

Thanks for your valuable advice. Changes have been made to some section of the manuscript.

**2.7.** In general, details are too vague, specific values instead of terms like relatively slow (line 180) or $Q_{max}$ would help improve the clarity of the paper.

Yes sure, this specific values is irrelevant and has been deleted.

**2.8.** Seasonal rainfall totals, anticipated rainfall totals, and a catalog of past floods in the city would all be helpful to give the reader more context.

This mean is covered in our previous article, which is currently under review. Thanks for your valuable advice.

**2.9.** There are several unpublished references, but surely there are published ones that can be used. Also, why are they unpublished (in review, etc.)

This manuscript is second of the previous article. The previous article is under review. I mean that.

**2.10.** Section 5.1: How were hazardous and vulnerable areas delineated? Are the maximum inundation depths known in the depressions? Without the paper's results being applied to a numerical model, it is hard to fully understand the scope of the hazard and the area of the city exposed to the hazard.

This is covered in section 4.2 of the manuscript, please see there. Maybe I'm wrong, please tell me again

**2.11.** Section 5.2: How did you come to these recommendations? Are they based on a similar case? Given the unique nature of the flood hazard in a city where 72% of the people reside in nomadic dwellings and are therefore extremely vulnerable, it would be helpful to understand why you think these points would be successful.

Mongolian traditional nomadic dwellings is a ger. The ger is very resistant to flooding. In this regard, it is based on a questionary survey of the local residents of Khovd City. Elderly residents of the flood-affected areas also spoke about this.

**2.12.** In the conclusion section, I don't think it is appropriate to say that a detailed flood hazard assessment was conducted. There is very little hydraulic data provided and the assessment that has been done is based exclusively on topographical surveys.

Yes sure, we chaged assessment to mapping…

**2.13.** In my opinion, a detailed flood hazard assessment would include modeling results where actual scenarios are tested that could delineate flooded zones and flood severity based on past and anticipated discharge values.

This mean is covered in our previous article, which is currently under review. Thanks for your valuable advice.

**Technical Corrections:**

**2.14.** Line 13: during *the* Summer

Corrected

**2.15.** Line 20: Change govern to governments or practitioners (or something similar)

The sentences were deleted

**2.16.** Line 21: Change on to in

The sentences were deleted

**2.17.** Line 22: Maybe change along basins to within basins?

Corrected
**2.18.** Lines 27-28: Suggest changing heavy and downpouring to extreme (or intense) rainfall events
Changed
**2.19.** Line 34: In the parentheses, is this a current emerging disaster caused by recent flooding or the 2016 and 2020 events?
This is a separate flood that occurred in the region bordering Khovd province.
**2.20.** Lines 35-36: I am not sure I would describe a situation with 10 floods occurring over 30 years as highly susceptible, especially compared to the flood frequencies of other mountain regions worldwide.
The Khovd city is located in a semi-arid region and on gulliy of lower part of high mountains, so it is very vulnerable to flash floods.
**2.21.** Line 48: change to expansions
Corrected
**2.22.** Line 51: delete a (…through a detailed topographical survey…)
Corrected
**2.23.** Line 82: $km^2$
Corrected
**2.24.** Line 83: I don't think gully is the appropriate term here also is flash flooding in this context only overland flow i.e. not channelized?
**2.25.** Or describe how these processes (flash vs river flooding) are different as they relate to the region i.e. flood flood-generating mechanism, orographic influences, hazard potential, etc.
**2.26.** Line 84: What is a yellow water flood?
Explained (snow meltwater)
**2.27.** Line 86: the river *is* covered
Corrected
**2.28.** Line 88: What is the $Q_{max}$ and how much has it increased? Add values, please.
Deleted
**2.29.** Line 110: How were the interviews used in the study?
We collected questionnaires from 54 local residents during the field mapping study and double-checked flood risk areas. Added a small sentence about it on 3.1.section. It Included:
*In addition, during the field mapping step, the flood hazard areas were checked under questionnaires from local residents.*
**2.30.** Line 115: What was the associated error of the validation process (relative and absolute accuracy)?
**2.31.** Line 124 (Figure 3) hiishade to hillshade.
Corrected
**2.32.** Line 128: What is (2022) referencing?
Deleted
**2.33.** Line 138: How were the 2 DEMs integrated to develop a flood hazard map?
You're right, the Alos Palsar DEM was not used in this study to a particularly high importance, so the all means was removed from the manuscript altogether. It was my mistake sorry.
**2.34.** Line 141: hiishade to hillshade
Corrected
**2.35.** Line 144: References in chronological order
**2.36.** Line 145: Is the newly developed DEM a combination of the 2 DEMs of unequal resolutions? More detail on this, please.
It developed by UAV only
**2.37.** Lines 148-155: Background information should be in the introduction.
Moved to the Study area
**2.38.** Line 160: Add space we developed

Corrected

**2.39.** Line 174: …it is located…

Corrected

**2.40.** Lines 179-180: not clear

Improved the sentence.

**2.41.** Line 182 Fig. 4: The red line is not in the legend (same in fig 1).

Added a explain sentence in fig 1 and there have red line on the legend of fig 4 which are two longitudinal profiles.

**2.42.** Line 190: why and how were cells grouped?

In detailing the DEM we created from the drone mapping and aerial images , we calculated the percentage of each cell showing its own flow direction into the atribute database on GIS. This is important in determining where surface runoff may accumulate and pose a hazard.

**2.43.** Line 203: References in chronological order.

It is automatically by Zotero

**2.44.** Line 217: What sediment analyses were executed? Please provide details as to the sampling and analysis methodology.

Sorry it was mistake, we couldn't take sediment sample, and it is just sediment character and feature so I we changed the word for mean

**2.45.** Line 221: Sustained

Corrected

**2.46.** Lines 223-224: Are these based on the interviews shown in Fig. 2. If so add a reference.

Corrected

**2.47.** Line 231: Severe?

Corrected

**2.48.** Lines 234-243: All background information for the introduction. Most of section 4.3 is background information.

In this manuscript, I wanted to provide an understanding of the risk of flash floods in the dry valley and gullies of Khovd city and the flood risk of the Buyant River. These 2 flood conditions are separate understands and conditions. In other words, this city is located in the middle of the flash flood condition and the river flood condition.

**2.46.** Figures 5-7: Missing red dashed line in legend.

Added explain sentence in the figure desription

*The red dashed line shows the profile slope.*

---

## Author Response (AR2)

Review to Flood hazard mapping and disaster prevention recommendations based on detailed topographical analysis in Khovd City, Western Mongolia, Author(s): Serdyanjiv Narangerel et al., MS No.: nhess-2024-91

**Report #2**

Anonymous referee #4, 04 Jan 2025

Dear Reviewer (Anonymous referee #4, 04 Jan 2025)

First of all, I would like to sincerely thank you for your valuable comments and suggestions. I have revised the manuscript thoroughly in accordance with your feedback. I paid particular attention to enhancing the scientific and methodological contributions to flood mapping. Specifically, I have made revisions to the sections on the Introduction (1), Area of Study (2), Photogrammetry by Drone (3.1), and Results (4), with corrections to sentence structure and meaning. Additionally, I rewrote the sections on Discussion and Conclusions in line with your comments.

All changes are highlighted in red in the manuscript.

I look forward to your reply.

Thank you once again.

---

## Author Response (AR4)

**Flood Hazard Mapping and Disaster Prevention Recommendations Based on Detailed Topographical Analysis in Khovd City, Western Mongolia, Author (s): Narangerel Serdyanjiv et al., MS No.: nhess-2024-91**

**Report#1**
**Submitted on 12 Sep 2025**
**Anonymous referee #5**

First of all, thank you very much for all your valuable comments and suggestions. Please see below for the changes made based on your feedback. The revised sections are highlighted in red.

**Comments:**
1. With the inclusion of the "recommendations for disaster prevention" the title now better fits to the manuscript. However, I would try to link the recommendations more clearly to the findings of the analysis, rather than giving very general recommendations.
2. L15 remove "such as Asian countries"
   L 15: Removed the words….
3. L19/20 and L22 Remove the abbreviation in the brackets (not necessary in the abstract, but introduce in the main text.)
   L 19/20 and 22: Removed the abbreviations in the brackets….
4. L31 Remove "for example,"
   L 31: Removed the word….
5. L64 these authors -> the authors
6. L 64: Changed the article….
7. L330 change "Key recommendations include i) enhancing flood protection facilities, ii) regular warnings to residents in flood-prone regions or river valleys, and iii) the implementation of local disaster management initiatives. Additionally, efforts should be made to improve the quality of weather prevention information, organize training sessions, and raise awareness about natural disasters among the citizenry." to "Key recommendations are i) enhancing flood protection facilities, ii) providing regular warnings to residents in flood-prone regions or river valleys, and iii) implementing local disaster management initiatives. Additionally, efforts should be made to improve the quality of weather prevention information, to organize training sessions, and raise awareness about natural disasters among the citizens."
   L 330: Changed the sentences….
8. - Figure 1: change caption to "Geographical settings and location of the study area of Khovd aimag in Western Mongolia (a, b). The red area shows the exact study area, including a western (c) and eastern (d) side. Photo of Khovd City, highlighting features such as the Buyant River floodplain, branches, streams (e), and the Khovd City area (e-Sentinel 2 satellite image)."
   Figure 1: Changed the sentences….